# Nanomedicines for Overcoming Cancer Drug Resistance

**DOI:** 10.3390/pharmaceutics14081606

**Published:** 2022-08-01

**Authors:** Tingting Hu, Hanlin Gong, Jiayue Xu, Yuan Huang, Fengbo Wu, Zhiyao He

**Affiliations:** 1Department of Pharmacy, State Key Laboratory of Biotherapy and Cancer Center, National Clinical Research Center for Geriatrics, West China Hospital, Sichuan University, Chengdu 610041, China; hxyyhtt@163.com (T.H.); leadsay@163.com (J.X.); wanzi916@wchscu.cn (Y.H.); 2Department of Integrated Traditional Chinese and Western Medicine, West China Hospital, Sichuan University, Chengdu 610041, China; 13880912113@163.com; 3Key Laboratory of Drug-Targeting and Drug Delivery System of the Education Ministry, Sichuan Engineering Laboratory for Plant-Sourced Drug and Sichuan Research Center for Drug Precision Industrial Technology, West China School of Pharmacy, Sichuan University, Chengdu 610041, China

**Keywords:** nanotechnology, nanomedicine, drug resistance, chemotherapy, targeted therapy, immunotherapy

## Abstract

Clinically, cancer drug resistance to chemotherapy, targeted therapy or immunotherapy remains the main impediment towards curative cancer therapy, which leads directly to treatment failure along with extended hospital stays, increased medical costs and high mortality. Therefore, increasing attention has been paid to nanotechnology-based delivery systems for overcoming drug resistance in cancer. In this respect, novel tumor-targeting nanomedicines offer fairly effective therapeutic strategies for surmounting the various limitations of chemotherapy, targeted therapy and immunotherapy, enabling more precise cancer treatment, more convenient monitoring of treatment agents, as well as surmounting cancer drug resistance, including multidrug resistance (MDR). Nanotechnology-based delivery systems, including liposomes, polymer micelles, nanoparticles (NPs), and DNA nanostructures, enable a large number of properly designed therapeutic nanomedicines. In this paper, we review the different mechanisms of cancer drug resistance to chemotherapy, targeted therapy and immunotherapy, and discuss the latest developments in nanomedicines for overcoming cancer drug resistance.

## 1. Introduction

Chemotherapy, targeted therapy and immunotherapy are the mainstream of cancer therapy; however, the most important challenge in cancer treatment is the development of cancer drug resistance [1,2,3]. Depending on the factors behind it, cancer drug resistance can be divided into primary drug resistance and acquired drug resistance [4,5,6]. Primary drug resistance refers to a complete failure of response to initial antitumor therapy in patients with cancer [7]. Acquired resistance is defined as a cancer that initially responds to antitumor therapy but then relapses or progresses after a period of treatment [8]. The mechanisms of cancer drug resistance are very complex including increased drug outflow, reduced drug intake, drug inactivation, target mutations, altered signaling pathway, defective apoptosis, phenotypic transition, autophagy, gene mutations or amplification, T cell depletion, and antigen deletion [9,10,11,12,13,14]. With the further study of drug resistance mechanisms, cancer stem cells (CSCs) and the tumor microenvironment are also closely related to the development of cancer drug resistance [15,16,17,18]. Therefore, there is a pressing need to develop new methods to surmount cancer drug resistance [19,20,21]. In recent decades, nanomedicines have emerged as promising cancer treatment tools, focusing on the targeted delivery of antitumor agents to tumor site using nanosized formulations such as polymeric micelles, liposomes, polymeric nanoparticles, inorganic nanoparticles, hybrid nanoparticles, and polymeric nanogels. Liposomal nanoparticles have the advantages of good biocompatibility, a long circulation time, tumor targeting ability and controlled release and have become among the most promising drug delivery platforms. Several nanomedicines, including liposome products, are in clinical use (Table 1). Additionally, a variety of liposomal nanoparticles have been widely studied for overcoming cancer drug resistance, especially breast cancer. Since most anticancer drugs are hydrophobic, if injected directly into the circulation, they not only have low bioavailability, but also have great toxicity. To address this conundrum, biodegradable amphiphilic block copolymers have become an attractive carrier material, because they can encapsulate or conjugate anticancer agents and then self-assemble into stable core–shell and nanostructured particles called polymeric micelles or polymeric capsules. Polymeric nanogels are formed with an internal network structure through intermolecular or intramolecular crosslinking, which can enhance the solubility of anticancer drugs by encapsulating hydrophobic or hydrophilic anticancer drugs inside the crosslinking networks. In addition, they have shown delayed drug release or rapid drug release based on the crosslinker degradation in response to the tumor microenvironment (TME). A variety of polymeric nanoplatforms have been widely studied for overcoming cancer drug resistance, including breast cancer, ovarian cancer, pancreatic cancer, and glioblastoma. Inorganic nanoparticles, including gold nanoparticles and silicon nanoparticles, have become a research hotspot in imaging and drug delivery platforms due to their unique physical and chemical properties, easy fabrication, stability and high drug loading ability. A variety of inorganic nanoparticles have been widely studied for overcoming cancer drug resistance, especially breast cancer and hepatocellular carcinoma. Hybrid nanoparticles, including biocompatible polymer-coated inorganic nanoparticles, phospholipid-coated polymeric nanoparticles and biomimetic cell membrane-coated nanoparticles have the advantages of improved biocompatibility, enhanced stability, enhanced drug encapsulation and activated target capacity. A variety of hybrid nanoparticles have been widely studied for overcoming cancer drug resistance, including breast cancer, ovarian cancer, lung cancer, and melanoma. Nanomedicines have therefore been introduced as reliable strategies to improve therapeutic effectiveness, reduce harmful adverse reactions, as well as surmount cancer drug resistance [22,23,24,25]. Nanomedicines have paved the way for effective treatment of cancer by rationally designing strategies such as passive targeted drug delivery, active targeted drug delivery, co-delivery of combinatorial agents and multimodal combination therapy, and have broad prospects in overcoming drug resistance [26,27,28,29,30,31]. In this paper, we review the different mechanisms of cancer drug resistance to chemotherapy, targeted therapy and immunotherapy, and focus on the latest progress in nanomedicines for overcoming cancer drug resistance.

## 2. Nanomedicines against Drug Resistance in Chemotherapy

### 2.1. Mechanisms in Drug Resistance of Chemotherapy

Nowadays, chemotherapy is still the most widely used strategy for treating cancer; however, the biggest obstacle to this traditional strategy is the development of cancer drug resistance [43,44,45]. The mechanisms of drug resistance to chemotherapy are extremely complex [45] (Figure 1). Generally, the emergence of chemoresistance may be classified by the following pathways: (1) increased drug efflux by ATP-dependent pumps mediated by transmembrane transporters of the ATP-binding cassette (ABC) superfamily [46,47,48]; (2) reduced drug uptake mediated by altering specific cellular targets [49,50,51]; (3) inactivation of apoptotic pathways mediated by high expression of the Bcl-2 antiapoptotic family such as Bcl-2, Mcl-1 and Bcl-XL, which are mainly responsible for the reason why cancer cells can resist apoptosis [52,53,54]; (4) enhanced DNA repair ability that can contribute to the resistance of cancer by promoting genomic instability and mutation [55,56,57]; (5) alterations in specific drug targets [58,59]; (6) increased drug detoxification mediated by metabolism or biotransformation [11,60]. All in all, these resistance mechanisms can allow cancer cells to survive by easily changing different pathways, and ultimately resulting in chemotherapeutic failure.

### 2.2. Nanomedicines to Overcome Chemotherapy Resistance

Considering that chemotherapy resistance-related drug efflux proteins mainly reside in the nuclear membranes and blood, but not in the mitochondria [61,62], delivering chemotherapy agents into the mitochondria is an emerging strategy to surmount drug resistance to chemotherapy [63,64,65,66,67,68,69]. Yu et al. [70] constructed a weak acid-activated, charge-reversible, triphenylphosphonium (TPP)-based, “shell–core” nanosystem (DOX-PLGA/CPT/PD) for sequential facilitation of tumor accumulation, cellular uptake, mitochondria targeting, intracellular localization and surmounting drug resistance of MCF-7/ADR breast cancer (Figure 2a). Firstly, positively charged mitochondrial-targeting lipid-polymer hybrid nanoparticles (PLGA/CPT) were prepared from PLGA and C_18_-PEG_2000_-TPP (CPT) [71]. Then, DOX was loaded into PLGA/CPT nanoparticles to obtain DOX-PLGA/CPT. Lastly, positively charged PEI-DMMA (PD) shell was wrapped on the surface of positively charged DOX-PLGA/CPT to obtain negatively charged DOX-PLGA/CPT/PD with a diameter of ~150 nm. When DOX-PLGA/CPT/PD was treated at pH 6.5, the hydrolysis of amide in PD occurred, facilitating the elimination of electrostatic interaction between PLGA/CPT and PEI, ultimately resulting in the deshielding of PD to reveal DOX-PLGA/CPT and transformation of the charge from −24 to +19.2 mV (Figure 2b). Then, they studied the pharmacokinetics of DOX-PLGA/CPT/PD, and, the results showed that DOX-PLGA/CPT showed significantly slower clearance with a half-life time 15.84 h. After incubation with MCF-7/ADR cells at pH 6.5, DOX-PLGA/CPT/PD showed effective lysosome escape (Figure 2c), excellent mitochondrial-targeting capacity (Figure 2d) and superior cytotoxicity for overcoming DOX resistance by up-regulating the apoptosis-related proteins as well as down-regulating the antiapoptotic protein Bcl-2 (Figure 2e). Encouraged by the in vitro antitumor effect of DOX-PLGA/CPT/PD, Yu et al. evaluated the in vivo effect in MCF-7/ADR cell-bearing mice. The results show that DOX-PLGA/CPT/PD showed the best inhibitory effect on tumor growth and exhibited the best treatment effect, with a tumor inhibition rate (TIR) of 84.9% with no obvious side effects (Figure 2f).

Studies show that the exposure of tumor cells to chemotherapy drugs can result in hypoxia-inducible factor-1 (HIF-1) activation and stabilization [72,73], where HIF-1 plays an important part in drug resistance by regulating multidrug resistance protein (MRP), P-glycoprotein (P-gp), Bcl-2, etc. [74,75,76]. Moreover, HIF-1 can up-regulate the level of glutathione, which can bind with heavy metal ions, including cisplatin [77,78]. Therefore, inhibiting HIF-1 pathways during chemotherapy might be a promising method to circumvent chemo-resistance [79,80,81,82,83]. Acriflavine (ACF), a potent HIF-1 inhibitor, has been proven to bind to HIF-1α and thereby impede HIF-1α/β dimerization [84,85], which can be a useful strategy for sensitization of chemotherapy. In this regard, Zhang et al. [86] developed a new type of microporous silica-based co-delivery system (PMONA) to reverse the acquired resistance to cisplatin. Firstly, cisplatin was loaded into the polymeric mPEG-silane functionalized mesoporous silica nanoparticles inner core by reverse microemulsion method, where polymeric mPEG-silane was applied to maintain stability during blood circulation. To achieve tumor-specific glutathione (GSH)-triggered drug release, tetrasulfide bond-bridged organosilica was integrated to obtain the nanoparticles. Finally, ACF was loaded into the inner area of mircopores by electrostatic interactions to obtain ACF-loaded nanoparticles with a diameter of ~45 nm. After internalization by cancer cells, the outer organosilica shell of PMONA could be degraded by intracellular GSH, resulting in nanoparticle disassembly, drug release and synergistic regulation of multiple cancer-related signaling pathways. As shown in an in vitro release experiment, cisplatin and ACF had faster and higher cumulative release rates in a medium containing 10 mM GSH than in a medium containing 10 μM GSH, which confirmed that the tetrasulfide bond in organosilica enabled GSH-responsive disassembly and drug release. After incubation with A459 cells, PMONA exhibited stronger cell cytotoxicity, induced more apoptosis than the single drug-loaded nanoparticles by suppressing HIF-1-related proteins and decreased the level of intracellular GSH. Inspired by the result that ACF strengthens the curative effect of cisplatin in vitro, Zhang et al. assessed the in vivo antitumor effect in A459 cell-bearing mice. The results indicated that PMONA showed the best inhibitory effect on tumor growth and exhibited the best therapeutic effect with limited side effects. Additionally, the immunohistochemical experiment showed that PMONA enhanced cell death and apoptosis in tumor tissues mainly by down-regulating the levels of P-gp, MRP2, HIF-1-activated glutamate-cysteine ligase modifier subunit (GCLM), vascular endothelial growth factor (VEGF) and cystine transporter (xCT). Taken together, these results confirmed that ACF could combat cisplatin-acquired resistance by inhibiting HIF-1 function.

Hyperthermia, a non-invasive treatment strategy, has shown a competitive advantage in reversing drug resistance in cancer by suppressing the expression of drug efflux transporters [87,88,89,90,91,92]. Therefore, hyperthermia combined with chemotherapy is a hopeful treatment strategy for overcoming chemotherapeutic resistance [93,94,95,96,97,98]. Huang et al. [99] constructed smart, thermoresponsive, pH low insertion peptide (pHLIP)-modified gold nanocages (DOX@pPGNCs) to realize synergistic thermo-chemotherapy and overcome chemotherapeutic resistance. Firstly, thermoresponsive poly (di (ethylene glycol) methyl ether methacrylate-co-oligo (ethylene glycol) methyl ether methacrylate) (PMEO_2_MA-OEGMA) polymer was anchored to gold nanocages to PMEO_2_MA-OEGMA-modified gold nanocages, where PMEO_2_MA-OEGMA served as a temperature-sensitive gate guard at a lower critical solution temperature of ca. 41.6 °C. In other words, the PMEO_2_MA-OEGMA chains extended under 41.6 °C, sealing the pore of gold nanocages to prevent the leakage of drug into the blood; however, once the temperature increased up to 41.6 °C due to the NIR-induced photothermal effects, its chains shrunk, leading to opening of the pores of gold nanocages and fast DOX release (Figure 3a). Then, pHLIP was used to decorate PMEO_2_MA-OEGMA-modified gold nanocages to obtain pPGNCs, where pHLIP was a good candidate to enhance cancer cell internalization by conformational transition at the weakly acidic tumor microenvironment. Lastly, DOX was loaded into pPGNCs to obtain DOX@pPGNCs with a diameter of ~160 nm and a zeta potential of approximately −20 mV. As shown in Figure 3b, PMEO_2_MA-OEGMA was thermosensitive with a lower critical solution temperature of ca. 41.6 °C. In vitro release experiments indicated that the cumulative release of DOX increased from 3.7 to 20.1% after 5 min of NIR irradiation. More importantly, the rapid release of DOX was consistent under NIR irradiation in another cycle (Figure 3c), indicating that PMEO_2_MA-OEGMA was a very responsible gatekeeper to precisely control NIR-triggered DOX release from DOX@pPGNCs. Cytotoxicity experiments showed that the antiproliferation ability against MCF-7/ADR cells was strongest in the DOX@pPGNCs and NIR irradiation group at pH 6.5 (Figure 3d), suggesting that pHLIP could enhance cellular uptake of DOX@pPGNCs under a weak acid tumor microenvironment, and, upon NIR irradiation, DOX@pPGNCs could efficiently achieve synergistic thermo-chemotherapy to overcome cancer resistance. In vivo biodistribution experiments showed that DOX accumulation in tumor site of tumor-bearing mice treated with DOX@ pPGNCs and NIR irradiation was highest (Figure 3e), confirming that NIR irradiation-triggered photothermal effects of gold nanocages could further strengthen DOX accumulation. Inspired by the above experimental results, Huang et al. further assessed the in vivo treatment effect in MCF-7/ADR cell-bearing mice. The results indicated that DOX@pPGNCs achieved the strongest antitumor efficacy with a TIR of 97.3% (Figure 3f,g), indicating the highly effective synergistic thermo-chemotherapy in MCF-7/ADR cell-bearing mice.

**Figure 3 pharmaceutics-14-01606-f003:**
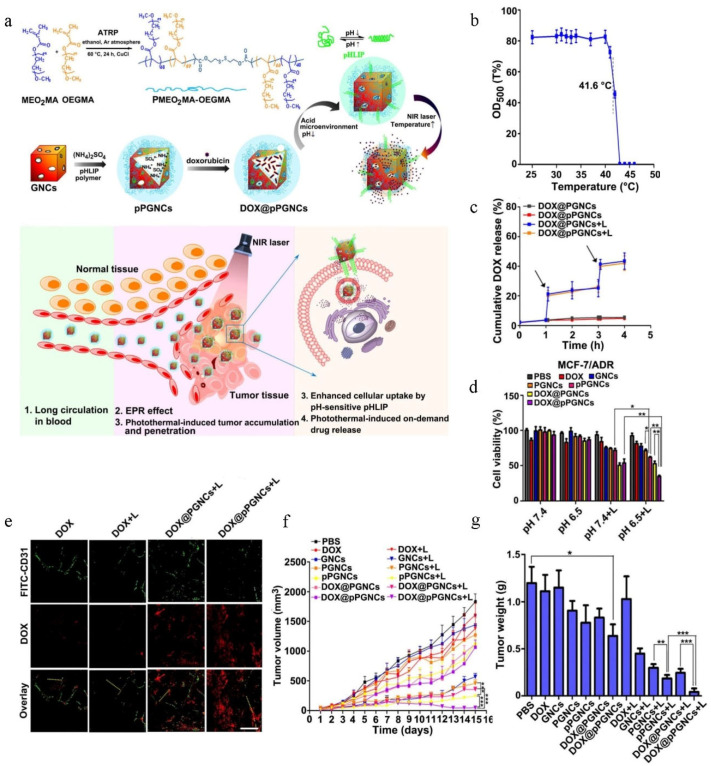
DOX@pPGNCs for chemo-photodynamic combination therapy of drug-resistant cancer. Reprinted with permission from Ref. [99]. 2019, NIH. (**a**) Schematic diagram of the mechanism of DOX@pPGNCs surmounting cancer drug resistance. (**b**) Transmittance of POEG in PBS. (**c**) In vitro release curve of DOX@pPGNCs. (**d**) In vitro cytotoxicity against MCF-7/ADR cells. ** p* < 0.05, ** *p* < 0.01. (**e**) Accumulation of different groups in tumor site of tumor-bearing mice. (**f**,**g**) In vivo antitumor effect of DOX@pPGNCs. ***
*p* < 0.05, ** *p* < 0.01, *** *p* < 0.001. P-gp, an ATP-dependent protein, is overexpressed in different drug-resistant tumor cells and closely related to chemotherapy resistance by promoting drug efflux [100,101,102,103,104]. In recent few years, RNA interference (RNAi) techniques have been used to block the expression of P-gp to reverse drug resistance [105,106,107,108,109]. Zheng et al. [109] constructed a special, siRNA and chemotherapy drug co-delivery system, an siRNA-based nanostructure (siRNAsome), to enhance combination therapy for overcome chemotherapeutic resistance (Figure 4a). Dynamic light scattering (DLS) data show that Dox.HCl-loaded siRNAsome (Pgp-siRNAsome@Dox.HCl) ranged in particle size from 126 to 135 nm. This novel siRNAsome was based on the self-assembly of siRNA-disulfide-poly (N-isopropylacrylamide) (siRNA-SS-PNIPAM) copolymers, which was very different from traditional siRNA delivery systems. In other words, this distinctive siRNAsome not only possessed an empty aqueous interior that could load hydrophilic agents, but also possessed a thermoresponsive and intracellular reduction-dependent hydrophobic median layer that could load hydrophobic drugs. Moreover, this siRNAsome possessed a siRNA stabilization shell that could load siRNA drugs without using a cationic component. When siRNAsome was incubated with a dithiothreitol (DTT) solution, DOX was rapidly released from the nanostructure, and more than 75% of the encapsulated DOX was released after 24 h incubation with dithiothreitol (Figure 4b), indicating that siRNAsome was sensitive to an intracellular environment and intracellular redox conditions could effectively disintegrate the structure of siRNAsome to control drug release. To test the capacity to efficiently deliver siRNA into tumor cells without the aid of a cationic component, MCF-7/ADR cells were incubated with siRNAsome and confocal laser scanning microscopy showed that siRNAsome could unquestionably promote uptake of siRNAsome (Figure 4c). More importantly, when treated with siRNAsome for 2 days, the P-gp mRNA level of MCF-7/ADR cells decreased by approximately 42% by P-gp gene silencing (Figure 4d). To test the synergistic antitumor effect of the siRNAsome, DOX and anti-P-gp siRNA were co-loaded into the siRNAsome to form Pgp-siRNAsome@Dox.HCl. As shown in Figure 4e,f, Pgp-siRNAsome@Dox.HCl showed the strongest cytotoxicity in MCF-7/ADR cancer cells and the strongest antitumor effect in MCF-7/ADR cell-bearing mice, indicating that the knockdown of P-gp mRNA could remarkably improve the activity of DOX to efficiently realize synergistic therapeutic efficacy and this cation-free Pgp-siRNAsome@Dox.HCl nanostructure could serve as a promising vehicle for reversing drug resistance.

Studies have shown that tumor cells can develop drug resistance by enhancing DNA repair [110,111,112,113,114,115], suggesting that drug resistance owing to DNA repair can be overcome by inhibiting the function of related proteins [116,117,118,119]. Recently, Wang et al. [120] constructed a smart delivery system for overcoming cisplatin-related “cascade drug resistance” (CDR) by mild hyperthermia (43 °C) triggered by NIR. Firstly, hydrophobic photothermal-conjugated polymer and biodegradable amphiphilic polymer were mixed to form F-nanoparticles (F-NPs) with photothermal performance. Secondly, biodegradable amphiphilic polymer and C16-CisPt-Suc (a Pt (IV) prodrug) were mixed to form Pt-nanoparticles (Pt-NPs). Lastly, Pt-NPs and F-NPs were mixed to obtain the mixed nanoparticles (F-Pt-NPs). On the basis of DLS data, the average particle size of F-NPs was 91.0 ± 2.6 nm, while that of the Pt-NPs was 105.1 ± 1.6 nm. In vitro experiments showed that, under the treatment of NIR, mild hyperthermia could efficiently facilitate cellular uptake of drug-resistant A549DDP cells, resulting in enhanced cytotoxicity and surmounting CDR of cisplatin by the consumption of GSH and the reduction of Pt (IV) to Pt (II). More importantly, mild hyperthermia could accelerate the binding of Pt to DNA and promote the formation of irreparable crosslinking of Pt-DNA strands, resulting in the destruction of DNA repair. In vivo experiments showed that, under mild hyperthermia conditions, F-Pt-NPs exhibited the best antitumor effect with a TIR of 94% with few side effects, further indicating that NIR-triggered mild hyperthermia could reverse CDR.

In recent years, substantial evidence has confirmed that drug resistance is closely related to the CSC phenotype [121,122]. One proven mechanism of multidrug resistance (MDR) in CSC is the increased expression of ABC transporters [123]. In addition, the CSC phenotype shows increased drug resistance to chemotherapy by modulating many other stem characteristics, including enhanced DNA damage repair capacity and up-regulation of antiapoptotic proteins [124,125]. Therefore, eradication of CSCs is an effective strategy to surmounting cancer drug resistance. Shen et al. [126] constructed an alltrans-retinoic acid (ATRA) and camptothecin (CPT) co-loaded nanoplatform (ATRA/CPT-NPs) to surmount chemotherapeutic resistance of both CSCs and bulk tumor cells. Firstly, ROS-responsive nitroimidazole-modified hyaluronic acid-oxalate-CPT conjugate (n-HA-oxa-CPT) was synthesized. Then, n-HA-oxa-CPT assembled into nanoparticles and physically encapsulated ATRA to obtain ATRA/CPT-NPs with a diameter of ~150 nm. Based on the difference levels of ROS between bulk tumor cells and CSCs, ATRA/CPT-NPS could sequentially release ATRA and CPT during the differentiation of CSCs. After uptake by hypoxia CSCs, ATRA was firstly released, which induced CSC differentiation into reduced stemness and chemoresistance, along with increased ROS level. Then, the increased ROS in differentiated CSCs triggered CPT release for enhanced cytotoxicity towards the differentiated cells with decreased drug resistance. On the other hand, after uptake by bulk tumor cells with hypoxia and high ROS, ATRA/CPT-NPS could simultaneously release ATRA and CPT, resulting powerful synergistic anticancer effects. In their study, ATRA/CPT-NPs showed the strongest inhibition efficacy on the orthotopic BCSC-enriched tumor mouse models, suggesting that the differential drug release realized by ATRA/CPT-NPs was very important to strengthen the synergistic efficacy of ATRA-triggered CSC differentiation and CPT-triggered cytotoxic activity for the treatment of poorly differentiated and highly chemo-resistant heterogeneous tumors.

Additionally, a summary of nanomedicines studied for overcoming chemotherapeutic resistance in recent years is displayed in Table 2.

## 3. Nanomedicines against Drug Resistance in Targeted Therapy

### 3.1. Mechanisms in Drug Resistance of Targeted Therapy

Compared with chemotherapy, targeted therapies exhibit superiorities in efficacy and safety, and is thus becoming the mainstream of clinical cancer therapy. Although great strides have made in the past two decades, there are still insurmountable problems that targeted therapy is facing [154,155,156,157,158,159,160]. The biggest problem is drug resistance [161,162]. Almost all targeted therapies encounter drug resistance after use for a period of time, which is mainly related to gene mutations, gene amplification, CSCs, efflux pumps, inactivation of apoptosis and autophagy [161] (Figure 5). Gene mutations are primarily responsible for resistance to targeted anticancer agents [163,164,165]—one argument is that the gene mutations are induced by drugs, and another is that drug-resistant mutations already existed. Gene amplification is another common cause of resistance associated with targeted anticancer agents [166,167]. According to statistics, approximately 20% of EGFR inhibitor-resistant cases are MET amplification. CSCs are also considered to be in charge of drug resistance and tumor recurrence due to their quiescence, epithelial-to-mesenchymal transition, resistance to DNA damage-induced apoptosis and interactions with the tumor microenvironment [168,169,170]. Overexpression of ATP-dependent efflux transporters, especially P-gp, also plays an important role in the drug resistance of targeted anticancer agents [171]. In addition to the resistance mechanisms mentioned above, autophagy [172] and inactivation of apoptosis [173] are also considered be responsible for drug resistance to targeted anticancer agents.

### 3.2. Nanomedicines to Overcome Targeted Therapy Resistance

AZD9291 (Osimertinib) is the first third-generation growth factor tyrosine kinase inhibitor (EGFR-TKI) approved by the FDA, which has significant effects not only on classical EGFR-sensitive mutations, but also on EGFR-resistant mutations (T790M mutation) [174,175,176]. Even so, it inevitably develops acquired drug resistance and this greatly impedes its benefits [177,178,179,180]. Studies have shown that autophagy allows cancer cells to improve their survival at advanced stages or under therapeutic stress [181,182,183], which ultimately results in small-molecule targeted anticancer drug resistance. On the other hand, fibroblast growth factor receptor 1 (FGFR1) is commonly expressed in all types of cancer and is involved in tumor progression and epithelial-to-mesenchymal transition-related drug resistance to EGFR-TKIs [184,185,186,187]. Thinking together, inhibition of both FGFR1 and autophagy might be an effective method to surmount AZD9291 resistance. Gu et al. [188] constructed a PD173074 (a kind of FGFR1 inhibitor) and chloroquine (CQ) dual-loaded shell–core targeted delivery system (CP@NP-cRGD), which could observably inhibit FGFR1 and autophagy, respectively, to reverse AZD9291 resistance (Figure 6a). Briefly, Gu et al. used DSPE-PEG and DSPE-PEG-cRGD to entrap PD173074 to obtain an interior core, and then utilized cRGD-PEG to load calcium phosphate (CaP) with CQ by a biomineralization method to form a shell. The average size of CP@NP-cRGD was 123.4 ± 0.4 nm, with a zeta potential of −15.1 ± 1.4 mV. CP@NP-cRGD could enhance cellular uptake by active targeting ligand cRGD and the pH-susceptive CaP shell could facilitate effective lysosome escape, thus promoting the successive release of CQ and PD173074. In vitro release experiments showed that CP@NP-cRGD was pH sensitive, where PD173074 exhibited a relatively sustained release behavior, whereas CQ showed a biphasic release behavior with an initial blasting effect and following sustained release (Figure 6b). When incubated with resistant cells, Rhodamine B (RB)-loaded CP@NP-cRGD showed the strongest fluorescence intensity (Figure 6c), indicating that cRGD could significantly promote cellular uptake of integrin αvβ3/αvβ5-rich cancer cells. Under the acidic conditions within lysosomes, the CaP shell of CP@NP-cRGD cracked, ion pairs formed, and the osmotic pressure of lysosomes increased, ultimately leading to lysosome cracking, indicating that CP@NP-cRGD could successfully facilitate lysosome escape to ensure effective drug delivery into the cytoplasm (Figure 6d). In vitro cytotoxicity testing showed that the CP@NP-cRGD and AZD9291 group demonstrated the strongest antiproliferative effect on resistant cells. Flow cytometry analysis and Western blotting tests further confirmed that CP@NP-cRGD and AZD9291 induced the highest rate of apoptosis, the most G0/G1 cell cycle arrest and reduced proliferation of resistant cells by down-regulation of p-ERK and up-regulation of cleaved-caspase 3 (Figure 6e). Moreover, Gu et al. confirmed the inhibitory effect of CP@NP-cRGD on autophagy. As shown in Figure 6f, CP@NP-cRGD and AZD9291 could inhibit autophagosome fusion with lysosomes, leading to the most remarkable accumulation of autophagosomes and the strongest LC3-II levels. Together, these results indicated that CP@NP-cRGD could reverse drug resistance to AZD9291. Inspired by the in vitro results, the in vivo anticancer effect of CP@NP-cRGD was tested in AZD9291-resistant NSCLC xenograft-bearing mice. Compared with other treatment groups, the CP@NP-cRGD and AZD9291 group exhibited the slowest tumor growth curves and showed the strongest antitumor effect (Figure 6g), suggesting that CP@NP-cRGD could effectively surmount drug resistance to AZD9291 in vivo. In addition, the effect of CP@NP-cRGD on apoptosis and autophagy in vivo was also studied. The results indicated that CP@NP-cRGD could efficaciously up-regulate the level of apoptosis-related protein cleaved-caspase 3 and down-regulate the level of proliferation-related protein Ki-67, which were highly consistent with the in vitro results. Autophagosome changes were further tested in tumor biopsies and the results revealed that the CP@NP-cRGD and AZD9291 group showed the most significant accumulation of autophagosomes as with the in vitro results. To sum up, CP@NP-cRGD could observably decrease proliferation and trigger apoptosis by impeding autophagy and the FGFR1 pathway in drug-resistant tumor, which provided a new strategy for reversing AZD9291 resistance.

Hypoxia is the most common pathological phenomena owing to the abnormal blood vessels and rapid cell proliferation at tumor site [189,190,191]. Studies have confirmed that hypoxic tumor cells overexpress HIF-1α and P-gp, which observably promote the progress of cancer cells resistant to small-molecule targeted drugs [192,193,194]. Therefore, there is a pressing need for an effective hypoxia-targeting strategy [195]. Bera et al. [196] constructed a kind of hypoxia-responsive copolymer (CMP-HA-NI-PEI-NBA) containing nitroaromatic subunits, which could self-assemble into nanomicelles to efficiently deliver erlotinib into cancer cells. The average size of erlotinib-loaded nanomicelles was 755.77 ± 51.11 nm, with a zeta potential of 26.62 ± 2.20 mV. After selective bioreduction of the nitroaromatic residues and rapid decomposition of the nanostructures in cancer cells deprived of oxygen, erlotinib could be rapidly released and eventually improve its chemo-sensitivity. When incubated with PBS and sodium dithionite (pH 5.0), erlotinib-loaded nanomicelles were depolymerized, leading to significantly rapid drug release behavior as compared to normal oxygen states. As compared to incubation at 4 °C, cellular uptake was significantly enhanced when incubated at 37 °C, indicating that cellular uptake of erlotinib-loaded nanomicelles was energy dependent. However, cellular uptake was significantly blocked after incubation of the cells with β-cyclodextrin, indicating the mechanism of caveolae/lipid raft-mediated endocytosis for the internalization of nanomicelles. In addition, erlotinib-loaded nanomicelles could trigger more apoptotic cells than erlotinib. Further, interestingly, erlotinib-loaded nanomicelles achieved significantly stronger cytotoxicity under hypoxia conditions than in normoxic conditions, owing to their hypoxia-dependent accelerated drug release.

To date, combinations of chemotherapy and light-mediated therapy have been investigated to surmount cancer drug resistance [197,198,199,200,201]. Sonodynamic therapy (SDT), an alternative treatment strategy, has been shown to be more effective than photodynamic therapy (PDT) because ultrasound penetrates soft tissue to a depth of approximately 10 cm [202,203,204,205,206]. However, the oxygen consumption of SDT increases the hypoxia degree of the tumor tissue, which in turn impedes the efficiency of SDT [207,208,209]. Considering that hypoxia is closely related to drug resistance of EGFR-TKIs, with oxygen being a pivotal substrate of SDT, Zhang et al. [210] developed a nanoplatform (namely CEPH) prepared by erlotinib-conjugated chitosan, which collaboratively delivered sonosensitizer hematoporphyrin (HP) and oxygen storage agent perfluorooctyl bromide (PFOB) to enhance the effect of SDT and overcome EGFR-TKI resistance. The particle size of CEPH was 284.1 nm and the zeta potential of CEPH was −8.4 mV. In their study, CEPH was sensitive to the acidic tumor environment and revealed a rapid release behavior in acidic condition. Moreover, the HP release rate of CEPH was significantly increased after ultrasound irradiation, and was higher than that of CEPH with no irradiation. These results suggested that CEPH was a pH/ultrasound dual-dependent delivery carrier. Under hypoxia conditions, cellular uptake of H1975 cells incubated with CEPH was higher than when incubated with CEH, indicating that PFOB could alleviate the hypoxia state and thus improve the cellular endocytosis of cancer cells with EGFR^T790M^ resistant mutation. In addition, compared with other nanocarriers, the toxicity of CEPH to H1975 cells was significantly enhanced, especially under hypoxic conditions, indicating that erlotinib and PFOB could collectively improve the antitumor effect. Moreover, CEPH and ultrasound showed the strongest antitumor effect due to the synergistic effects of erlotinib and SDT. The above results confirmed that erlotinib, PFOB and HP could generate synergistic effects to improve the antitumor effect and reverse EGFR-TKI resistance by combination of erlotinib and SDT. A three-dimensional spherical model of multicellular tumor showed that the hypoxic condition was aggravated after treatment with ultrasound alone but relieved when treated with CEPH and ultrasound, indicating that SDT was an oxygen-consuming procedure but PFOB in CEPH could alleviate the hypoxia induced by SDT and enhance the SDT effect. Western blotting results further expounded that CEPH and ultrasound was able to observably inhibit the expression of EGFR, down-regulate the level of HIF-1α and increase the level of ROS, which ultimately contribute to the cooperative antitumor effect.

In addition to drug resistance, another big challenge that small-molecule targeted anticancer drugs are facing is metastases [211,212,213]. Studies have shown that approximately 33% of non-small-cell lung cancer (NSCLC) patients with EGFR mutation develop brain metastases (BMs) [214], which was the main cause of mortality in patients with advanced NSCLC. Thereby, there is an urgent need to design an effective strategy for surmounting drug-resistant brain metastases. Yin et al. [215] designed a dual-targeting drug delivery system (T12/P-Lipo) modified with T12 peptide and anti-PD-L1 nanobody for synergistic delivery of simvastatin/gefitinib to surmount the two huge obstacles in the treatment of NSCLC—drug resistance and brain metastases. Where T12 peptide could facilitate blood–brain barrier (BBB) penetration, anti-PD-L1 nanobody was used as a targeting ligand for active targeting and simvastatin was utilized to repolarize the tumor-associated macrophages from the M2 to M1 phenotype for remodeling tumor microenvironments. The average size of the T12/P-Lipo was approximately 153 nm, with a zeta potential of approximately −27 mV. A Transwell culture model was utilized to assess the advantage of T12/P-Lipo in BBB penetration. When incubated with H1975 cells, the levels of transferrin receptor and PD-L1 in the capillary endothelial cells (BCECs) were significantly increased (Figure 7a), suggesting that both transferrin receptor (TfR) and PD-L1 were the promising target receptors in the brain metastases vascular endothelial cells. Moreover, cellular uptake of T12/P-Lipo by H1975 cells was higher than that of any other nanocarriers (Figure 7b), and the fluorescence images further verified the intensive BBB-penetrating capacity of T12/P-Lipo and enhanced cellular uptake (Figure 7c). After incubation with human umbilical vein endothelial cells, simvastatin and gefitinib exhibited significant down-regulation of CD206 (an M2-related marker) and up-regulation of STAT1 and iNOS (an M1-related marker) (Figure 7d), indicating that the combination of simvastatin and gefitinib could effectively repolarize macrophages from M2 to M1. An in vitro cytotoxicity study was performed in EGFR^T790M^ H1975 cells. After treatment, T12/P-Lipo showed the maximum antiproliferation ability with IC_50_ 1.85 μg/mL. Western blotting further expounded that T12/P-Lipo up-regulated the expression of nicotinamide adenine dinucleotide phosphate oxidases 3 (NOX3), down-regulated the levels of methionine sulfoxide reductase (MsrA), antioxidant enzyme glutathione peroxidase 4 (GPX4) and Bcl-2 (Figure 7e), suppressed EGFR/Erk/Akt phosphorylation (Figure 7f) and thereby triggered increased cleaved caspase-3 to reverse drug resistance (Figure 7g). Then, they studied the pharmacokinetics of T12/P-Lipo, and, the results showed that the half-life time of T12/P-Lipo was 5.9 h. Inspired by the in vitro experiments, in vivo antitumor efficacy was evaluated in the BM mouse model. As shown in Figure 7h,i, the antitumor effect of T12/P-Lipo was better than that of other groups, indicating the prospect of dual-targeting drug delivery in improving the treatment of BMs and surmounting the cancer resistance of EGFR^T790M^ mutation. In addition, Western blotting demonstrated that T12/P-Lipo had a strong inhibitory effect on EGFR/Erk/Akt tumor growth signaling pathway, which was highly consistent with the in vitro results. Additionally, the T12/P-Lipo treatment group induced extensive apoptosis of the BMs by down-regulating caspase 3 and Ki 67. To sum up, T12/P-Lipo could efficiently dual-target the brain metastases and tumor-associated macrophages, providing a novel strategy for the treatment of advanced NSCLC with drug-resistant brain metastases.

Additionally, a summary of nanomedicines studied for overcoming resistance to targeted therapy in recent years is displayed in Table 3.

## 4. Nanomedicines against Drug Resistance in Immunotherapy

### 4.1. Mechanisms in Drug Resistance of Immunotherapy

The emergence of immunotherapy, such as immune checkpoint inhibitors (ICIs) and adoptive cell metastases (ACT), has changed the pattern of traditional tumor surgery, radiotherapy and chemotherapy and is becoming an important method of clinical tumor treatment, especially hematologic cancers and solid tumors [222,223,224,225]. However, the clinical curative effect of these immunotherapies is restrained by cancer drug resistance, which has become among the important challenges in immunotherapy [226,227,228]. The mechanism of immunotherapy resistance is completely different from that of traditional chemotherapy drug resistance, mainly due to the “loss or deficiency” of antigens [229,230]. An abnormal signal pathway, loss or mutation of tumor suppressor genes, loss or down-regulation of target antigen, JAK1/2, B2M and Apelin receptor gene mutation, TIM/LAG3 expression, inhibitory checkpoint increase, activation checkpoint decrease, T cell depletion, phenotypic changes, tumor microenvironment and drug antigenicity all play important roles in immunotherapy resistance [231,232,233,234,235,236,237,238].

### 4.2. Nanomedicines to Overcome Immunotherapy Resistance

Recent studies have shown that gene-encoding phosphatase and tensin homolog deleted on chromosome 10 (PTEN) is directly related to the regulation of antitumor immunity. Firstly, there is a significant correlation between loss of PTEN and reduced T cell infiltration, which eventually leads to resistance to PD-1 monoclonal antibody [239,240,241]. In addition, the disfunction of PTEN also promotes the aggregation of suppressive immune cells, such as bone marrow-derived suppressive immune cells (MDSCs) and regulatory T cells (Tregs) [242,243,244,245]. More importantly, the loss of PTEN expression has been shown to down-regulate autophagy [246,247], which can effectively support tumor development. Therefore, Lin et al. [248] constructed a PTEN mRNA nanocarrier (mPTEN@NPs) delivered to tumor cells with PTEN deletion or mutation to restore the immunosuppressive TME, stimulate immune response and improve the efficacy of immune checkpoint blockade (ICB) therapy by inducing autophagy activation and damage-associated molecular patterns (DAMPs) release. The particle size of mPTEN@NPs was 111.8 ± 15.3 nm. In their study, mPTEN@NPs could restore tumor sensitivity to immunotherapy, as well as trigger the release of DAMPs and the function of autophagy, which thereby promoted the formation of autophagosomes. In vivo studies showed that PTEN repair induced a strong response of CD8+ T cells and restored TME by inhibiting the generation of Tregs and monocyte MDSCs and promoting the generation of pro-inflammatory cytokines. In addition, they evaluated the antitumor effect of mPTEN@NPs combined with anti-PD-1 immunotherapy in tumor models with PTEN deletion or mutation, suggesting that this combination therapy strategy had significant therapeutic efficacy and immune memory. All above results suggested that repairing tumor suppressors with mRNA nanomedicine could improve the sensitivity of tumors to ICB therapy and provided an effective cooperative therapy strategy for a variety of malignancies.

Skin is an immune organ containing a large number of resident antigen-presenting cells [249,250,251]. Microneedles can penetrate the immune cell-rich epidermis and induce a strong immune response by stimulating the function T cells [252,253]. In recent years, researchers have used microneedles as an adjunctive immunotherapy for melanoma [254,255,256,257]. Lan et al. [258] constructed a microneedle patch containing pH-dependent tumor-targeting nanoparticles (aPD-1/CDDP@NPs) that could synergistically deliver cisplatin (CDDP) and aPD-1locally to tumor tissues for enhanced antitumor efficacy (Figure 8a). First of all, aPD-1/CDDP@NPs were developed by a reverse-phase microemulsion method. Then, aPD-1/CDDP@NPs were wrapped in dissolved microneedles by a molding method to form aPD-1/CDDP@NPs MNs, which could facilitate immune regions on the skin via transdermal delivery. As shown in Figure 8b, an MN was composed of 9 × 9 needles, where the needle height was 588 μm and the basal diameter was approximately 240 μm. The fluorescent image of the aPD-1/CDDP@NPs MNs also demonstrated the sufficient distribution of aPD-1/CDDP@NPs on the tips of MNs (Figure 8c). The obtained aPD-1/CDDP@NPs MNs could be sufficiently dissolved in water within 5 min and penetrated thoroughly into mouse skin within 20 min. In vivo antitumor results showed that the animal model in their study did not respond to routine aPD-1 immunotherapy. Nevertheless, aPD-1 delivered by MNs showed a remarkable antitumor effect. The tumor weight treated by aPD-1 MNs (0.05 ± 0.017 g) was 8-fold lower than that treated by routine aPD-1 (0.443 ± 0.083 g). More importantly, the tumor regression effect was most significant in the apD-1/CDDP@NP MN group, with a tumor weight of 0.012 ± 0.005 g (Figure 8d). The above results indicated that aPD-1 delivered by MNs could obtain effective anticancer effects in animal models that did not respond to routine aPD-1 immunotherapy, and collaborative delivery of aPD-1 and CDDP by MNs could achieve stronger antitumor effects. To assess T-cell responses, blood was harvested and analyzed. As shown in Figure 8e,f, the positive rates of CD4+ T cells and CD8+ T cells treated by aPD-1 were only 15.0% and 12.13%, while the positive rates of CD4+ T cells and CD8+ T cells in the aPD-1 MN group were 37.50% and 47.98%, respectively. More importantly, aPD-1/CDDP@NP MNs had the highest T-cell infiltration rate (75.95% of CD8+ T cells) of any groups. ELISA results further indicated that the production of IFN-γ increased after aPD-1 nanoencapsulation. Notably, MN-mediated drug delivery promoted greater IFN-γ expression, and the level of IFN-γ was the highest when treated by aPD-1/CDDP@NP MNs (Figure 8g). In addition, they analyzed the infiltration of CD4+Foxp3+ T cells and found a significant decrease in Tregs in the three MN groups (Figure 8h). These results further verified that aPD-1/CDDP@NP MNs could trigger the activity of T cells, as well as kill tumor cells through T cells.

In recent years, natural killer (NK) cell-related immunotherapy has been revealed as an alternative to ICB-based immunotherapy or vaccine-based immunotherapy [259,260,261,262]. However, its therapeutic effect is largely limited by down-regulation of the recognition ligands, and its immunological effect can further be blocked by the secretion of the tumor microenvironment such as transforming growth factor-β (TGF-β) [263,264,265]. Researchers have confirmed the advantages of methods that combine immunotherapy with chemotherapy in the treatment of cancers [266,267,268]. Therefore, there is great interest in developing combined strategies to improve NK cell immunity. Selenium (Se) is a key trace element that plays a crucial role in cancer prevention and immune response. Selenium can promote the generation of CD8+ T cells and increase the secretion of perforin and granulation enzyme [269]. In addition, Se can enhance the immunologic function of adoptive NK cells through TNF-related apoptosis [270]. Therefore, Se-based drugs combined with immunosuppressive ablators can be beneficial to improve the immune activity of NK cells. Liu et al. [271] constructed a nanoemulsion delivery system (SSB NMs) to synergistically load selenocysteine (SeC) and TGF-β inhibitors to realize an amplified immunotherapeutic effect. The particle size of SSB NMs was approximately 130 nm and the zeta potential of SSB NMs was approximately −5.32 mV. In their study, the cytotoxicity of NK92 cells to HCC1937 cells and MMDA-MB-468 cells were significantly increased by SSB NMs pretreatment for 12 h. Moreover, activated NK cells obtained from patients were further used to evaluate the immunosensitization triggered by SSB NMs. Pre-incubation of target cells could improve the lysis efficiency of NK cells from 0.94 to 13.8 with an average increase of 0.2 to 30.6 folds. These results confirmed that SSB NMs could trigger the immunosensitization properties of NK92 cells. Mechanism studies shown that SSB NMs could effectually inhibit TGF-β/TGF-β RI/Smad2/3 pathway and enhance the expression of NKG2DL triggered by DNA destruction-dependent pathway, thus improving the efficacy of immunotherapy on NK92 cells. More importantly, after pretreatment of SSB NMs twice, adoptive therapy of NK92 cells showed the strongest antitumor effect with tumor inhibition rate of 78.15%. In conclusion, the above results demonstrated that the synergistic treatment therapy of SSB NMs and adoptive NK92 cells had a high efficacy in cancer treatment.

Although ICB therapy has greatly improved the survival of treatment-sensitive patients, the overall response rates to ICB therapy remains low [272,273,274], so, combination therapy with other immunomodulators that intercept non-redundant inhibitory pathways is being actively studied to surmount cancer drug resistance to ICB [275,276]. Since ICB are inherent to T cells, lymphocytes and not cancer cells are the target cells of ICB. Taken together, direct targeting of lymphocytes after administration has high prospects [277,278,279]. Francis et al. [280] designed an antibody-modified nanoplatform (ANCs) that could efficiently distribute in lymph nodes and lymphocyte-rich tissues and thus contributed to ICB-mediated antitumor effect. The average particle size of ANCs was approximately 30 nm. In their study, ANCs allowed for the encapsulation of TGF-β inhibitor or adenosine antagonists within the NPs core, while maintaining the binding affinity of surface-conjugated monoclonal antibodies to facilitate targeted delivery of encapsulated agents to lymph nodes or lymphocyte-rich tissues. Utilizing ICB monoclonal antibody as targeting ligand as well as signal-blocking therapy, ANCs slowed the tumor growth, prolonged the animal survival and thus improved the therapeutic effects of TGF-β receptor 1 inhibitor and adenosine 2A antagonist. Over all, the combination formulation strategy enabled multiple immunotherapy agents to be co-delivered to T lymphocytes and had the potential to strengthen immunotherapeutic effect by simultaneously inhibiting non-redundant inhibitory pathways.

Additionally, a summary of nanomedicines studied for overcoming resistance to immunotherapy in recent years is displayed in Table 4.

## 5. Discussion

Cancer is among the leading causes of death worldwide. Although chemotherapy, targeted therapy and immunotherapy have made significant achievements in the treatment of cancer, current treatment strategies all have serious limitations and often fail in clinical treatment. There are many reasons for treatment failure, such as poor oral bioavailability, and drug-related severe adverse reactions. The most challenging problem is that patients receiving antitumor therapy almost invariably develop drug resistance. In this respect, great progress has been made in uncovering the mechanisms of cancer drug resistance. However, how to overcome drug resistance of cancer is still an unsolved medical problem in clinical treatment. Therefore, cancer drug resistance has become a major obstacle in cancer treatment, and surmounting drug resistance is the goal oncologists are actively exploring now and in the future.

In recent years, nanomedicines, a new and effective strategy for surmounting cancer drug resistance, have attracted increasing attention. Nanomedicines can improve the bioavailability of difficult soluble drugs, extend drug blood circulation, pass through biological barriers, achieve targeted tumor therapy by passive or active targeting, improve the antitumor effect and use their own physical properties (optical, electric or magnetic) for real-time imaging, localization and control of drug release. In addition, nanomedicines can co-deliver therapeutic agent combinations for synergistic treatment or multimodal therapy, making them attractive therapeutic options for overcoming cancer drug resistance. As reviewed in this context, a variety of unique nanomedicines, including liposomes, polymer NPs and hybrid NPs, have been constructed to surmount cancer drug resistance. As more nanomedicines are developed and optimized, the advantages of nanomedicines over current treatment strategies will continue to be exploited. Therefore, it is believed that nanomedicines will be an attractive strategy for reversing or overcoming cancer drug resistance.

However, there is still a long way to go when currently studied nanomedicines can be used in clinic. First of all, due to the rapid development and wide application of nanomedicines or nanomaterials, the research on their potential toxicity has not been fully determined. Therefore, attention should be taken to develop nanomedicines with good safety characteristics to avoid damage from systematic exposure of the nanomaterials themselves, and, more in vivo safety and efficacy data are needed to support the clinical application of nanomedicines. Secondly, more effort should be made to collect clear evidence for the mechanisms of action, rather than to construct many new complex nanomedicines for similar concepts. Among the primary goals of nanomedicines is to ensure therapeutic agents to be delivered into tumor cells at sufficient local concentrations with minimal loss of their activity in the bloodstream. In addition, nanomedicines should reduce potential toxicity to normal tissues and healthy cells. Studies have shown that the composition, size, morphology and surface charge of nanomedicines can all affect their biological effect and distribution in organisms. How to design nanomedicines with a stable structure and good function is still a huge challenge. Nanomedicines larger than 150 nm mainly accumulate in spleen, liver and lungs. Nanomedicines under 5 nm mainly accumulate in kidney. Nanomedicines in between this range are more easily captured in the spleen and liver than in the lungs [289]. Spherical nanomedicines are mostly captured in the liver, while rod-shaped nanomedicines remain in the spleen and liver [290]. The smaller the cell–nanoparticle contact points, the more easily nanomedicines internalize into cancer cells. Thus, spherical and ovoid nanomedicines are more easily internalized than elongated nanomedicines [291]. Positively charged nanomedicines are more likely to be filtered out of blood by spleen, liver and lungs. Additionally, therefore, nanomedicines with a positive charge are the least suitable for maintaining prolonged blood circulation [292]. More importantly, the performance of nanomedicines can be improved by a variety of modifications, such as antibodies, peptides and cell membranes, to improve drug bioavailability, change surface charge, improve pharmacokinetic parameters or promote active targeting [293,294]. Lastly, another challenge that nanomedicines are facing is how to deliver therapeutic agents to defined cells. The difficulty is how to effectively identify the target cell among thousands of non-target cells. In this regard, active targeting nanomedicines offer fairly effective strategies [295]. The passive targeting of nanomedicines mainly depends on the enhanced permeability and retention (EPR) effect, and the distribution of nanomedicines largely relies on their size. Researchers proposed that the optimal size of nanoparticles is approximately 100 nm for long blood circulation and tumor accumulation [296]. However, active targeting nanomedicines can achieve this goal regardless of particle size. Active targeting refers to the modification of nanomedicines’ surface with ligands, such as peptides, antibodies, polysaccharides and aptamers, that can bind specifically to the target spots present on cancer cells, which can lead to a receptor- or antigen-mediated targeting strategy [297]. A large number of studies have shown that active targeting nanomedicines have more advantages than passive targeting nanomedicines. Therefore, active targeting nanomedicines are more suitable for drug delivery to overcome cancer drug resistance.

## 6. Conclusions

This review introduces the latest research progress of nanomedicines in overcoming cancer drug resistance. The study findings are very encouraging and strongly indicate that nanomedicines have great potential to overcome drug resistance in cancer. Moreover, this review discusses challenges and suggestions for the successful construction of nanomedicines with improved properties for overcoming cancer drug resistance, paving the way for their introduction into the clinic. Lastly, this review also provides a convenient guide for researchers to achieve a better understanding of significant findings in the field of nanomedicines to overcome cancer drug resistance experienced in recent years.

## Figures and Tables

**Figure 1 pharmaceutics-14-01606-f001:**
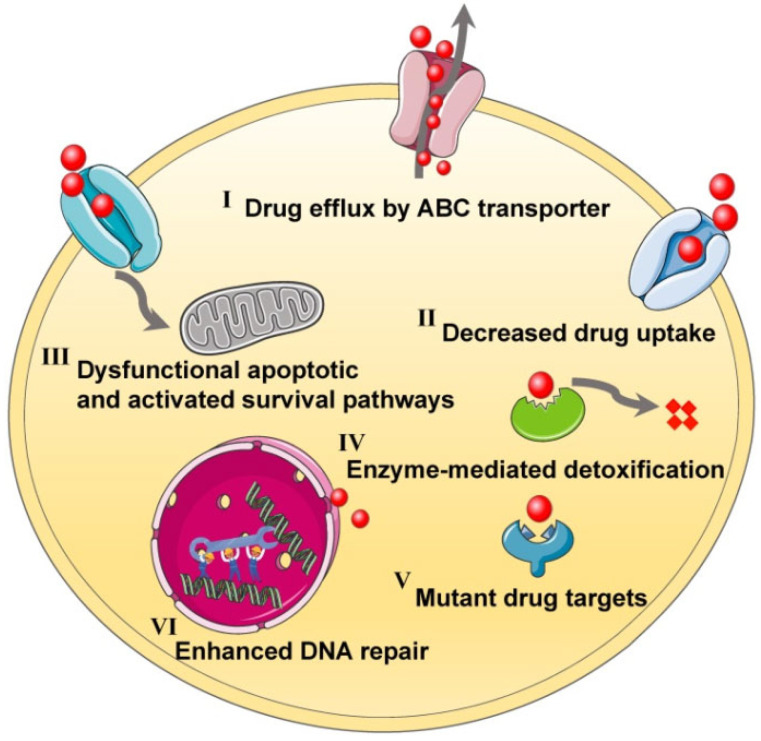
Mechanisms in drug resistance of chemotherapy. Reprinted with permission from Ref. [45]. 2021, NIH.

**Figure 2 pharmaceutics-14-01606-f002:**
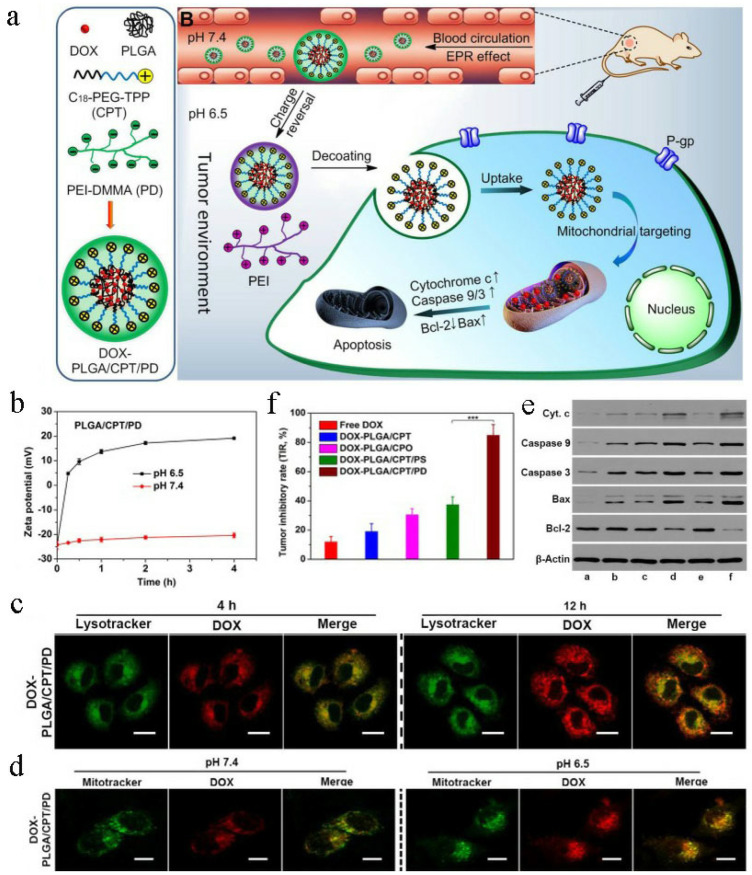
DOX-PLGA/CPT/PD for overcoming cancer drug resistance. Reprinted with permission from Ref. [70]. 2019, NIH. (**a**) Schematic diagram of the mechanism of DOX-PLGA/CPT/PD surmounting cancer drug resistance. (**b**) Charge reversal of PLGA/CPT/PD. (**c**) Lysosome escape of PLGA/CPT/PD, scale bar = 20 μm. (**d**) Mitochondrial targeting capability of DOX-PLGA/CPT/PD, scale bar = 20 μm. (**e**) Apoptosis mechanism analysis. (**f**) Tumor inhibition of DOX-PLGA/CPT/PD after treatment for 18 days.

**Figure 4 pharmaceutics-14-01606-f004:**
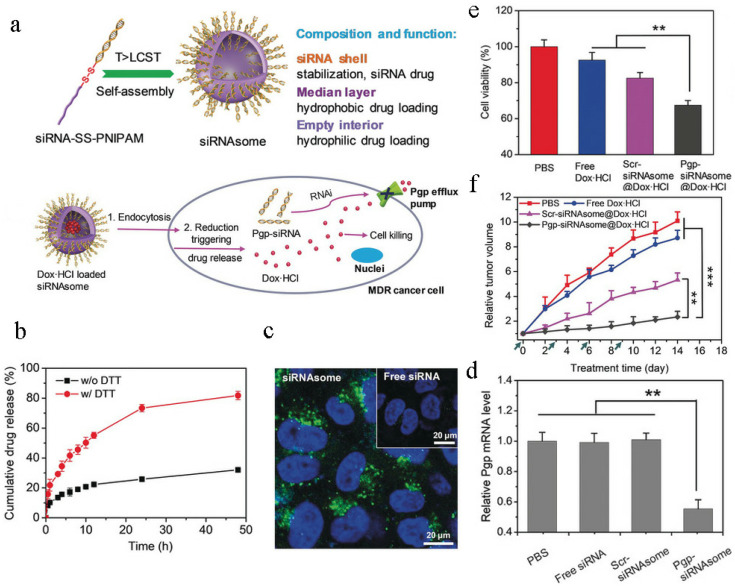
Pgp-siRNAsome@Dox.HCl for synergistic therapy against drug-resistant cancer. Reprinted with permission from Ref. [109]. 2019, Wiley. (**a**) Schematic diagram of the mechanism of Pgp-siRNAsome@Dox.HCl surmounting cancer drug resistance. (**b**) Dox.HCl release from Pgp-siRNAsome@Dox.HCl. (**c**) Cellular uptake of Pgp-siRNAsome@Dox.HCl. (**d**) Gene silencing of Pgp mRNA level in MDR MCF-7 cells. ** *p* < 0.01. (**e**) In vitro cytotoxicity against MDR MCF-7 cells. ** *p* < 0.01. (**f**) In vivo antitumor effect of Pgp-siRNAsome@Dox.HCl. ** *p* < 0.01, *** *p* < 0.001.

**Figure 5 pharmaceutics-14-01606-f005:**
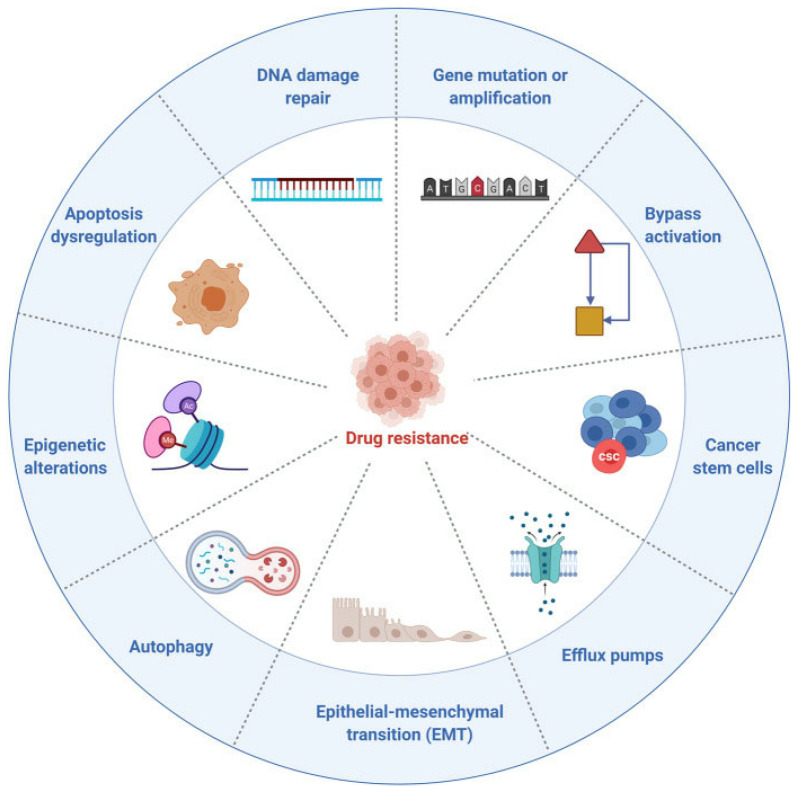
Mechanisms in drug resistance of targeted anticancer agents. Reprinted with permission from Ref. [161]. 2021, Springer.

**Figure 6 pharmaceutics-14-01606-f006:**
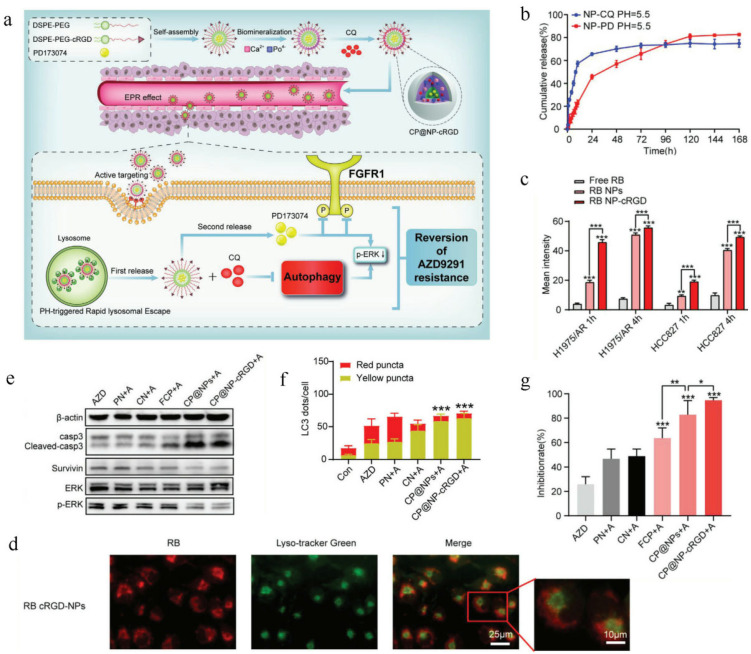
CP@NP-cRGD for overcoming cancer drug resistance. Reprinted with permission from Ref. [188]. 2020, Wiley. (**a**) Diagram of CP@NP-cRGD intracellular mechanisms of overcoming drug resistance. (**b**) In vitro release curve of CP@NP-cRGD. (**c**) Cellular uptake of RB-loaded CP@NP-cRGD in AZD9291-resistant cells. ** *p* < 0.01, *** *p* < 0.001. (**d**) The lysosomal escape of CP@NP-cRGD. (**e**) Antitumor molecular mechanisms of CP@NP-cRGD. (**f**) CP@NP-cRGD inhibited autophagy of H1975/AR cells. *** *p* < 0.001. (**g**) In vivo antitumor efficiency of CP@NP-cRGD. * *p* < 0.05, ** *p* < 0.01, *** *p* < 0.001.

**Figure 7 pharmaceutics-14-01606-f007:**
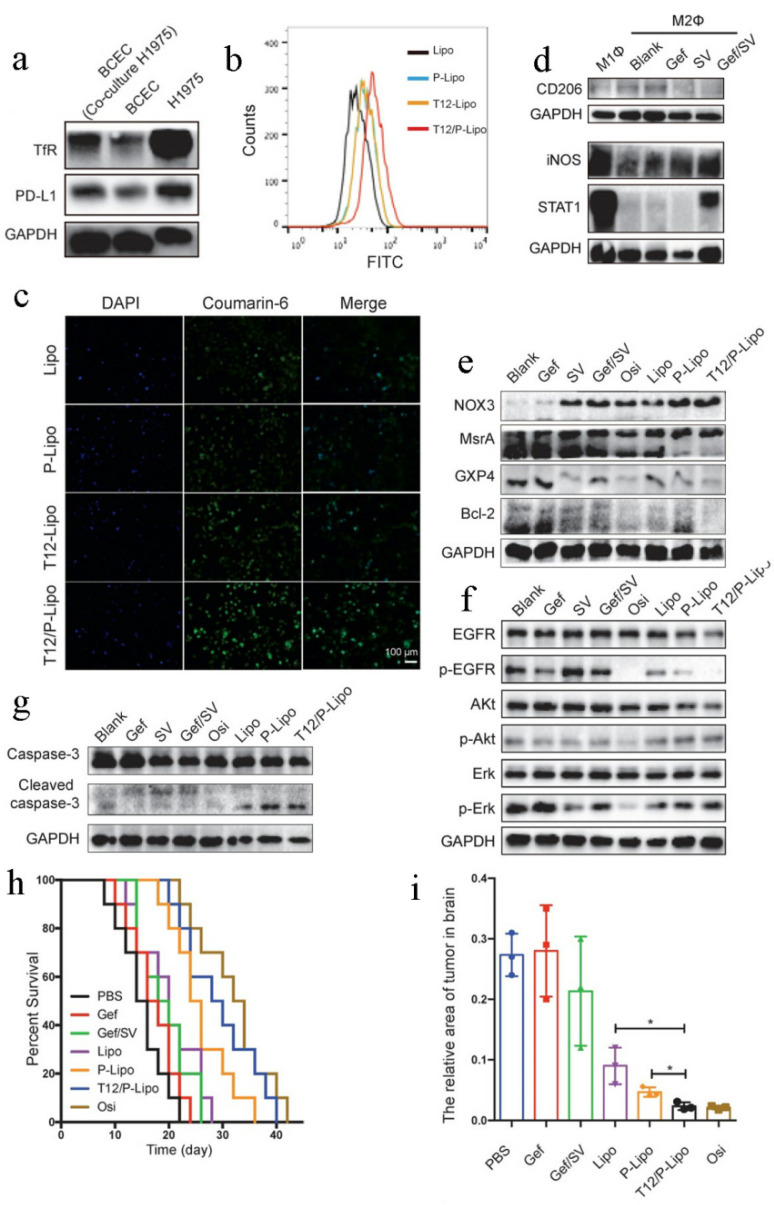
T12/P-Lipo for overcoming cancer drug resistance and brain metastases. Reprinted with permission from Ref. [215]. 2020, NIH. (**a**) The targeting receptors in BCECs. (**b**) Uptake of T12/P-Lipo by H1975 cells. (**c**) Fluorescence images of T12/P-Lipo in H1975 cells. (**d**) Macrophage repolarization from M2 to M1. (**e**) The level of NOX3/MsrA/GPX4/Bcl-2 in H1975 cells. (**f**) The level of phosphorylated EGFR/p-Akt/p-Erk. (**g**) The level of cleaved caspase 3. (**h**) The survival curves. (**i**) Tumor regression after treatment with T12/P-Lipo. * *p* < 0.05.

**Figure 8 pharmaceutics-14-01606-f008:**
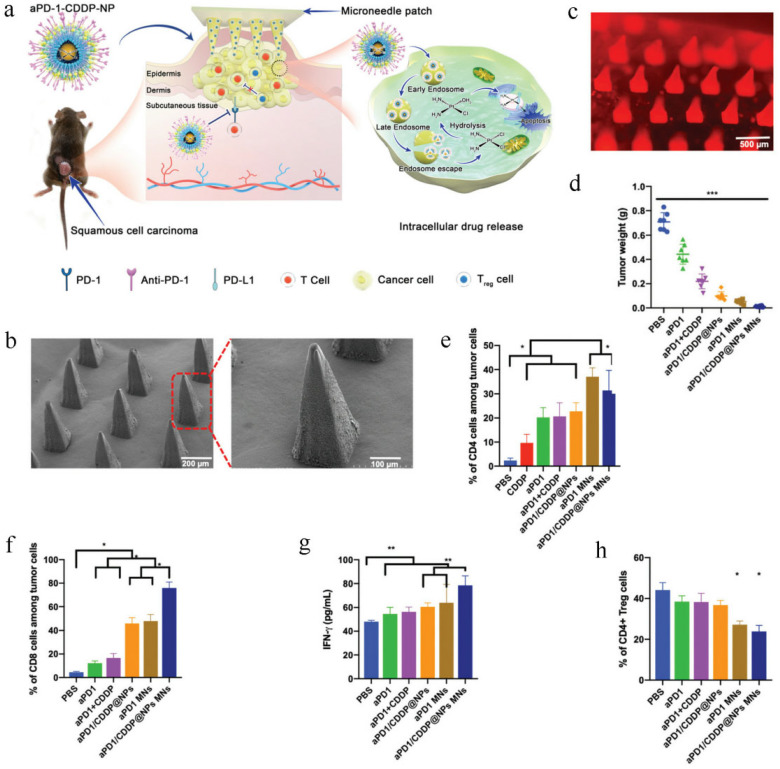
Schematic diagram of synergistic effect of immunochemotherapy by a microneedle patch containing aPD-1/CDDP@NPs. Reprinted with permission from Ref. [258]. 2020, Royal Society of Chemistry. (**a**) The mechanism of enhanced immunotherapy induced by aPD-1/CDDP@NP MNs. (**b**) SEM images of dissolving microneedles. (**c**) Fluorescence microscopy image of the microneedles. (**d**) Antitumor efficacy in vivo. *** *p* < 0.001. (**e**) The positive rate of CD4+ T cells. * *p* < 0.05. (**f**) The positive rate of CD8+ T cells. * *p* < 0.05. (**g**) IFN-γ expression in the serum. ** *p* < 0.01. (**h**) The percentages of CD4+FOXP3+ Tregs. * *p* < 0.05.

**Table 1 pharmaceutics-14-01606-t001:** Examples of the approved nanomedicines.

Trade Name	Active Ingredient	Nanoformulation	Indications	Approval Date	Clinical Effect	Reference
Doxil^®^	Doxorubicin	Liposome	Ovarian and breast cancer	1995	Fewer and less severe side effects Longer periods of drug circulation in vivo Prolonged interval to progression and progression-free survival time	[32]
DaunoXome^®^	Daunorubicin	Liposome	HIV-related Kaposi’s Sarcoma	1996	No obvious cardiotoxicity Longer periods of drug circulation in vivo As effective as the conventional chemotherapy	[33]
DepoCyt^®^	Cytarabine	Liposome	Lymphomatous meningitis	1999	Reduced cardiotoxicity Pronounced effectiveness	[34]
Eligard^®^	Leuprolide	Nanosphere	Prostate cancer	2002	More sustained testosterone suppression Higher response rates	[35]
Lipusu^®^	Paclitaxel	Liposome	Ovarian cancer and breast cancer	2003	Reduced adverse reactions As effective as the paclitaxel	[36]
Abraxane^®^	Paclitaxel	Nanoparticle	Metastatic breast cancer	2005	Lower overall toxicity Better anticancer effect	[37]
Genexol-PM^®^	Paclitaxel	Micelle	Breast cancer and Lung cancer	2007	Reduced toxicities of paclitaxel Enhanced antitumor efficiency	[38]
Marqibo^®^	Vincristine	Liposome	Acute lymphoid leukemia	2012	Reduced neurotoxicity Overall increase in therapeutic index	[39]
Onivyde^®^	Irinotecan	Liposome	Metastatic pancreatic cancer	2015	Longer half-life Enhanced anticancer efficiency	[40]
Liporaxel^®^	Paclitaxel	Emulsion	Gastric cancer	2016	Reduced neurotoxicity As effective as paclitaxel	[41]
Vyxeos^®^	Daunorubicin and cytarabine	Liposome	Acute myeloid leukemia	2017	Prolonged overall survival (OS) and event-free survival (EFS)	[42]

**Table 2 pharmaceutics-14-01606-t002:** Recent advances in nanomedicines for overcoming chemotherapeutic resistance.

Nanoformulation	Name	Particle Size	Payload	Reversal Mechanism of Drug Resistance	Cell Line	Tumor Model	Reference
Polymeric micelles	ACP-Dox and Apa micelles	104 ± 2 nm	DOX and apatinib	Inhibit P-gp activity	MCF-7/ADR cells	MCF-7/ADR tumor-bearing mice	[127]
	HA-PLGA (PTX and FAK siRNA)-NPs	232.9 ± 6.9 nm	PTX and FAK siRNA	siRNA-mediated silencing of FAK	HeyA8-MDR and SKOV3-TR cells	Drug-resistant, patient-derived xenograft (PDX) model	[128]
	ACP-R837 and PPP-DOX	~110 nm	R837 and DOX	Synergistic chemo-immunotherapy	4T1 cells	4T1 tumor-bearing mice	[129]
	NC-DOX	~122 nm	DOX and IR780	Combined chemotherapy/PTT/PDT	MCF-7/ADR cells	MCF-7/ADR tumor-bearing mice	[130]
Polymeric nanoparticles	Dox-Cur-NDs	55.1 ± 3.0 nm	DOX and CUR	Down-regulate the expression of P-gp	A2780 ADR cells	A2780 ADR tumor-bearing mice	[131]
	[FeFe]TPP/GEM/FCS NPs	176.0 ± 17.2 nm	Gemcitabine and [FeFe]TPP	Reduce the of function P-gp efflux pump	T24 cells	T24 tumor-bearing mice	[132]
	IGU-PLGA-NPs	199.6 nm	Iguratimod	Facilitate BBB penetration and inhibit GSCs proliferation and stemness	U87 and U251TMZ-R cells	U87 tumor-bearing mice	[133]
Liposomes	rTLM-PEG, PTX liposomes	/	PTX and trichosanthin	Reverse caspase 9 phosphorylation and induce caspase 3-dependent apoptosis	A549/T cells	A549/T tumor-bearing mice	[134]
	PTX/NO/DMA-L	146.3 ± 0.82 nm	PTX and DETA NONOate	NO-mediated down-regulation of P-gp	A549/T cells	A549/T tumor-bearing mice	[135]
	CBZ liposomes	108.53 ± 1.5 nm	CBZ	G2/M phase arrest	MCF-7 and MDA-MB-231 cells	Female SD rats	[136]
	Lip (Ap-Dox)	128.6 nm	Ap-Dox complex	Bypass the P-gp-mediated drug efflux	MCF-7/ADR cells	MCF-7/ADR tumor-bearing nude mice	[137]
	(DEX and DTX)-Lip	74.02 ± 0.41 nm	DTX and dexamethasone	Overcome stroma obstacles	Multidrug-resistant KBv cells and 4 T1 cells	Multidrug-resistant KBv and metastatic 4 T1 tumor models	[138]
	FPL-DOX/IM	159 ± 6 nm	DOX and imatinib	Inhibit ABC transporter function	MCF-7/ADR cells	MCF-7/ADR tumor-bearing mice	[139]
	PpIX/Dox liposomes	55.9 ± 20.9 nm	DOX and PpIX	Disrupt the structure of P-gp	MCF-7/ADR cells	MCF-7/ADR tumor-bearing mice	[140]
Nanogels	LNGs-PTX-siRNA	~100 nm	PTX and MDR1 siRNA	Knockdown MDR1	DROV cells	DROV tumor-bearing mice	[141]
	CDDP/DOX-NGs	~100 nm	CDDP and DOX	Combination chemotherapy	MCF-7/ADR cells	MCF-7/ADR tumor-bearing mice	[142]
	HA/Cis/Dox	45 ± 9.9 nm	DOX	GSH-induced DOX release	A2780cis cells	/	[143]
	SiPT75	75.5 ± 19.8 nm	TPPS	Elude the drug efflux pumps and retards exocytosis of cells	A549/DDP cells	A549/DDP tumor-bearing mice	[144]
Inorganic nanoparticles	H-MSNs-DOX/siRNA nanoparticles	~100 nm	P-gp siRNA and DOX	siRNA-mediated silencing of P-gp	MCF-7/ADR cells	MCF-7/ADR tumor-bearing mice	[145]
	Pt-AuNS	~85 nm	Pt	GSH depletion and GPX4 inactivation	MCF-7/ADR cells	MCF-7/ADR tumor-bearing mice	[146]
	FA-GT-MSNs@TPZ	~60 nm	TPZ	Synergistic radio-chemo-photothermal therapy	Hypoxic SMMC-7721 cells	SMMC-7721 tumor-bearing mice	[147]
Hybrid nanoparticles	S_CA4P_NP_BTZ_	~150 nm	BTZ and CA4P	Inhibit the overexpression of BCRP/ABCG2	A549 cells	Human A549 pulmonary adenocarcinoma xenograft model and PDX model of colon cancer	[148]
	cNPs	286 ± 79 nm	Afatinib, rapamycin and docetaxel	Synergistic treatment	HER2-positive breast cancer cells, EGFR-positive NSCLC cells and SKBR-3/AR cell lines	HER2-positive breast cancer mouse model	[149]
	4T1-HANG-GNR-DC	103.1 ± 7.6 nm	CDDP and DOX	Synergistic chemo-photothermal therapy	4T1 cells	4T1 tumor-bearing mice	[150]
	IR780/DTX-PCEC@RBC	~150 nm	IR780 and DTX	Combination therapy	MCF-7 cells	MCF-7 tumor-bearing mice	[151]
	cNC@PDA-PEG	170.5 ± 1.4 nm	Paclitaxel/lapatinib	Combination therapy	MCF-7/ADR cells	/	[152]
	miR497/TP-HENPs	125 ± 6 nm	miR497 and triptolide	Synergically suppress mTOR signaling pathway	SKOV3-CDDP cells	SKOV3-CDDP tumor-bearing mice	[153]

“/”: The original research article did not mention it.

**Table 3 pharmaceutics-14-01606-t003:** Recent advances in nanomedicines for overcoming resistance to targeted therapy.

Nanoformulation	Name	Particle Size	Payload	Reversal Mechanism of Drug Resistance	Cell Line	Tumor Model	Reference
Polymeric micelles	CP@NP-cRGD	123.4 ± 0.4 nm	CQ and PD173074	Dual FGFR1-autophagy blockade	H1975/AR and HCC827/AR cells	H1975/AR tumor-bearing mice	[188]
	CsA/Gef-NPs	37.1 ± 13.1 nm	Cyclosporin A and gefitinib	Cyclosporin A-mediated gefitinib sensitization	PC-9-GR cells	PC-9-GR tumor-bearing mice	[216]
Polymeric nanoparticles	ELTN and FDTN@PEG-PLA	~120 nm	Fedratinib and Erlotinib	Inhibit the JAK2/STAT3 signaling pathway	Erlotinib-resistant H1650 cells	Erlotinib-resistant H1650 xenograft tumor model	[217]
	CE7Ns	234.2 ± 8.5 nm	Cy7 and erlotinib	Synergistic erlotinib-targeted therapy and photodynamic therapy	PC-9 and Erlotinib-resistant H1975 cells	PC-9 tumor-bearing mice	[218]
	ERL-loaded CMP-HA-NI-PEI-NBA	755.77 ± 51.11 nm	Erlotinib	Hypoxia-triggered rapid drug release	Drug-resistant hypoxic HeLa cells	/	[196]
Liposomes	T12/P-Lipo	~153 nm	Simvastatin and gefitinib	TAM targeting and enhanced BBB penetration	EGFRT^790M^-mutated H1975 cells	EGFRT^790M^-mutated H1975 brain metastasis model	[215]
	P-Lipo	156 nm	Simvastatin and gefitinib	Neovascularization regulation and M2-macrophage repolarization	EGFRT^790M^-mutated H1975 cells	EGFRT^790M^-mutated H1975 tumor-bearing mice	[219]
	tLGV	~180 n	Gefitinib and vorinostat	TAM reprogramming	EGFRT^790M^-mutated H1975 cells	EGFRT^790M^-positive H1975 tumor model	[220]
Hybrid nanoparticles	ACLEP	184.8 ± 5.87 nm	Erlotinib and PFOB	Reverse hypoxia-induced drug resistance	A549 and Erlotinib-resistant H1975 cells	A549 tumor-bearing mice	[221]

“/”: The original research article did not mention it.

**Table 4 pharmaceutics-14-01606-t004:** Recent advances in nanomedicines for overcoming resistance to immunotherapy.

Nanoformulation	Name	Particle Size	Payload	Reversal Mechanism of Drug Resistance	Cell Line	Tumor Model	Reference
Hybrid nanoparticles	mPTEN@NPs	111.8 ± 15.3 nm	PTEN mRNA	Improve the sensitivity of ICB therapy	B16F10 cells	B16F10 melanoma tumor-bearing mice	[248]
	PGA@GOx@Mn, Cu-CDs	~80 nm	Gox and Mn, Cu-CDs	Combined action of starving therapy/PDT/PTT and checkpoint-blockade immunotherapy	4T1 cells	4T1 tumor-bearing mice	[281]
	BBPQDs	30 nm	BPQDs	Reprogram the immunosuppressive TME	4T1 cells	4T1 tumor-bearing mice	[282]
	R837@HM-NPs	71 ± 4.1 nm	R837	Reprogram the immunosuppressive TME	4T1 cells	4T1 tumor-bearing mice	[283]
	CAT@S/Ce6-CTPP/DPEG	~100 nm	Catalase and Ce6	Combined action of PDT and immunotherapy	4T1 cells	4T1 tumor-bearing mice	[284]
Liposomes	H_2_O_2_@Liposome and CAT@Liposome	~140 nm	H_2_O_2_ and Catalase	Radio-immunotherapy	4T1 cells	4T1 tumor-bearing mice	[285]
Nanoemulsion	SSB NMs	~130 nm	SeC and TGF-β inhibitor	Improve the sensitivity of cell-based immunotherapy	MDA-MB-231 cells	MDA-MB-231 tumor-bearing mice	[271]
Polymeric nanoparticles	ANCs	~30 nm	ICB mAb and small-molecule immunomodulators	T lymphocyte targeting and combination therapy	B16F10 and 4T1 cells	B16F10 and 4T1 tumor-bearing mice	[280]
Nanogels	P407 hydrogel	~28 nm	Anti-CTLA-4 antibodies	Sustained antibody release	D1DCs and MC-38 cells	CT26 tumor-bearing mice	[286]
	Zeb-aPD1-NPs-Gel	~100 nm	Zebularine and anti-PD1 antibody	Controlled drug release and reversal of immunosuppressive TME	B16F10 cells	B16F10 melanoma-bearing mice	[287]
	aPDL1-GEM@Gel	/	Gemcitabine and PD-L1 blocking antibody	Combination therapy	B16F10 and 4T1 cells	B16F10 and 4T1 tumor-bearing mice	[288]

“/”: The original research article did not mention it.

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
