# Peer review of "Nanomedicines for Overcoming Cancer Drug Resistance"

_pharmaceutics, 2022, doi:10.3390/pharmaceutics14081606_

Round 1

Reviewer 1 Report

In the article (pharmaceutics-1798816), the author made substantial research efforts to survey the literature and summarize the latest updates on the role of nanomedicine to treat drug resistance associated with cancer therapies. A detailed classification and explanation of the different cancer therapies (including Chemotherapy, targeted therapy and immunotherapy) and the associated drug resistance have been well described. The authors emphasized and illustrated the implication of selected recent nanomedicine in the alternation of cancer drug resistance. Finally, existing data on the possible role of this nanomedicine in developing potential therapies were compared and discussed. Overall, the surveyed literature and protocol applied to achieve its purpose are adequate, well-structured, well-presented, and actual. However, there are several recent reviews have been reported on this topic (e.g., 10.3390/cancers13092058, 10.1016/j.addr.2013.09.019 , 10.1016/j.drup.2021.100777). Further, the authors have not fully covered the latest updated findings on this topic. Accordingly, I would suggest that the authors put more effort into this article in order to make it more novel. I recommend that the manuscript be accepted for publication in Pharmaceutics journal after addressing the following revision and suggestions:

- the authors should modify the introduction part to better introduce the nanomedicine topic/types and shortly highlights their role in the different types of cancer.

- it is suggested that the authors include and cover the main findings from different sub-types of nanomedicine and highlight the effect/significant change in specific/certain types of nanomedicine instead of just talking generally.

- the authors should also cover the latest reported studies about cancer stem cells and drug resistance.

- the authors should add a separate section (Table) that summarizes the most common and known nanomedicine to target drug resistance (for each drug type) that has been previously applied.

- the authors should include in the conclusion part the outlook for therapeutic invention based on targeting cancer resistance and the currently known nanomedicine.

- the manuscript requires moderate language editing. The manuscript contains several grammatical mistakes that must be corrected.

Reviewer 2 Report

Comments: The work is an interesting topic,  However there are some conceptual errors in this article, Would you please clarify following arguments?

1-    The manuscript is missing a comprehensive discussion If some formulations have reached clinical application, the authors should state it clearly and discuss the clinical results achieved.

2-    Several parameter of nanoparticles (g. size, chemical composition and surface charge) that are important to understand their behavior in biological systems were not considered. Moreover, to highlight the putative benefits of targeted therapy using nanoparticles, the pharmacokinetics and pharmacodynamics parameters of nanoparticles encapsulated drugs should be presented.

3-    Nanoparticles used to overcome resistances of chemotherapy, targeted therapy and immunotherapy, should to be summarized in table with description of their ( structure, size, cell lines, animal model, mechanism of action, etc) and to be written down cross-ponding to each section.

4-     Authors should to highlight advantages and disadvantages of nanomedicine , as well as their potential side effects. Discuss why nanomedicines are advantageous in drug delivery, and how their  accumulation may depend upon the properties of the nanoparticles (e.g. size, surface charge, shape)

5-    Authors should also to provide comparison between active and passive targeting therapy and scheme  to illustrate their possible  mechanism as  delivery system.

Round 2

Reviewer 1 Report

the authors have addressed all concerns that have been raised and the manuscript has been significantly improved. Accordingly, I would recommend the publication of this interesting review article in the present form.

Author Response

Dear Professor,

We are very grateful for your satisfaction with our revised manuscript. Thank you again for your constructive comments which have helped us in depth to improve the quality of our manuscript.

Yours sincerely.

Reviewer 2 Report

Manuscript has been revised point by point according to reviewer comment. 

Manuscript is more acceptable NOW.

Author Response

(The authors gave the same response as above.)
